# Novel Isoform DTX3c Associates with UBE2N-UBA1 and Cdc48/p97 as Part of the EphB4 Degradation Complex Regulated by the Autocrine IGF-II/IR^A^ Signal in Malignant Mesothelioma

**DOI:** 10.3390/ijms24087380

**Published:** 2023-04-17

**Authors:** Pierluigi Scalia, Carmen Merali, Carlos Barrero, Antonio Suma, Vincenzo Carnevale, Salim Merali, Stephen J. Williams

**Affiliations:** 1The ISOPROG-Somatolink EPFP Research Network, Philadelphia, PA 19102, USA and 93100 Caltanissetta, Italy; 2Sbarro Institute for Cancer Research and Molecular Medicine and Center for Biotechnology, Temple University, Philadelphia, PA 19122, USA; 3Proteomics and Metabolomics Facility, Moulder Center for Drug Discovery Research, School of Pharmacy, Temple University, Philadelphia, PA 19140, USA; 4Institute of Computational Molecular Science, College of Science and Technology, Temple University, Philadelphia, PA 19122, USA

**Keywords:** IGF2, insulin-like growth factor-II, IR-A, insulin receptor isoform-A, EphB4, eph receptor B4, DTX3, Deltex 3, E3 ubiquitin ligase, UBE2N, E2 ubiquitin ligase, UBA1, E1 ubiquitin ligase, Cdc48/p97, chaperon protein also known as VCP

## Abstract

EphB4 angiogenic kinase over-expression in Mesothelioma cells relies upon a degradation rescue signal provided by autocrine IGF-II activation of Insulin Receptor A. However, the identity of the molecular machinery involved in EphB4 rapid degradation upon IGF-II signal deprivation are unknown. Using targeted proteomics, protein–protein interaction methods, PCR cloning, and 3D modeling approaches, we identified a novel ubiquitin E3 ligase complex recruited by the EphB4 C tail upon autocrine IGF-II signal deprivation. We show this complex to contain a previously unknown N-Terminal isoform of Deltex3 E3-Ub ligase (referred as “DTX3c”), along with UBA1(E1) and UBE2N(E2) ubiquitin ligases and the ATPase/unfoldase Cdc48/p97. Upon autocrine IGF-II neutralization in cultured MSTO211H (a Malignant Mesothelioma cell line that is highly responsive to the EphB4 degradation rescue IGF-II signal), the inter-molecular interactions between these factors were enhanced and their association with the EphB4 C-tail increased consistently with the previously described EphB4 degradation pattern. The ATPase/unfoldase activity of Cdc48/p97 was required for EphB4 recruitment. As compared to the previously known isoforms DTX3a and DTX3b, a 3D modeling analysis of the DTX3c Nt domain showed a unique 3D folding supporting isoform-specific biological function(s). We shed light on the molecular machinery associated with autocrine IGF-II regulation of oncogenic EphB4 kinase expression in a previously characterized IGF-II+/EphB4+ Mesothelioma cell line. The study provides early evidence for DTX3 Ub-E3 ligase involvement beyond the Notch signaling pathway.

## 1. Introduction

Destruction by the C-End (“Des-C-End”) is a ubiquitin-mediated mechanism targeting a subset of intracellular proteins aimed at proteolytic degradation [1] as part of a general protein homeostatic network [2]. The components of such molecular switches involve specific ubiquitin E3 ligases along with a defined and emerging set of E2 and E1 ubiquitin ligases according to each target [3,4]. Three families of ubiquitin E3 ligases have been found associated to “Des-C-end”, namely, Cullin-RING E3 ligases, HECT-RING E3 ligases, and C3HC4 RING E3 ligases [5,6,7]. Among these ubiquitin E3 ligase-driven complexes associated with protein C-terminal End-mediated degradation, the C3CH4-E3 ring-type complexes remain poorly characterized, as do their regulated cellular protein targets. DTX3 E3 Ligase is a representative of the C3CH4 ring group that includes DTX1, DTX2, and DTX3L; have been found to be involved in the regulation of Notch signaling [8,9,10,11,12,13]. EphB4 belongs to the family of EphB/Ephrin membrane tyrosine kinases [14,15], which is expressed in arterial endothelial cells during vasculogenesis [16,17,18,19] and has been found to be overexpressed in malignant cancer cells. It has been associated with a variety of cancer-promoting effects, including morphogenesis, angiogenesis, invasion, and metastasis [20,21,22,23,24,25]. We have previously found EphB4 protein levels, which are over-expressed in Malignant Mesothelioma cell lines recapitulating the three known histological types, to be tightly dependent on an intact autocrine IGF-II signal [26], with blocking of the secreted growth factor leading to a rapid and marked decrease in EphB4. Interestingly, such dependence on the IGF-II signal was more marked in the cell lines with higher malignant grades (MSTO211H and 2025H). We showed that EphB4 tyrosine phosphorylation on its C-terminal Degron/PDZ-binding domain by IGF-II was inversely related to its ubiquitination status. We further identified the EphB4 c-tail spanning its distal 30 amino acids as the phospho-inhibited degron carrying a UBX-like motif (aa 958–974) and phosphorylated by the IGF-II signal on EphB4 PDZ-binding motif (PQY) [27] at the level of Tyr987. These findings suggest that EphB4 protein integrity and ectopic expression in cancer depend upon an intact autocrine IGF-II steady state signal. However, the molecular players involved in such a mechanism are not yet known. In the present work, by means of ex vivo treatment and immune-enriched proteomics, a novel PTM-enhanced pull-down assay, co-immunoprecipitation, cDNA sequencing, and structural computational modelling uncover a complete set of ubiquitin ligases and a protein unfoldase/chaperone, and discloses them to be specific interactors of the recently identified EphB4 phospho-inhibited non-canonical c-Degron. Notably, this study discloses a novel DTX3 E3-Ligase isoform (DTX3c) as part of a specific EphB4 c-tail docking complex including the ubiquitin ligases UBA1(E1) [28] and UBE2N(E2) [29,30] along with the Cdc48/p97 triple ATPase/unfoldase [31]. The binding pattern of the factors identified with EphB4 herein is consistent with their direct involvement in the observed IGF-II/IR^A^ signal-mediated regulation of EphB4 protein levels in the studied cancer cell line.

## 2. Results

### 2.1. Identification of Selective Autocrine IGF-II Signal-Dependent Intracellular EphB4 Binders via Immuno-Affinity Proteomic Analysis

In order to identify specific intracellular mediators of the autocrine IGF-II rescue effect on rapid EphB4 degradation, we searched for autocrine IGF-II differentially regulated/enriched proteins in native EphB4 immunoprecipitates using LC-Ms/Ms profiling. Our search for IGF-II signal-dependent EphB4 interactors focused on MSTO211H cells among the three histological types recapitulating previously studied mesothelioma cell lines which displayed the highest dependence of EphB4 protein levels from the degradation rescue signal provided by IGF-II [26]. Considering the possibility of EphB4 ectoshedding mediated by ADAM proteins [32], which could limit EphB4 kinase recovery, and to enhance the identification of IGF-II regulated intracellular interactors, we tested (a) HUVEC cell lysates as a control source of native full length EphB4 interactors (both extracellular and intracellular) and(b) cell extracts from MSTO211H cultured in serum-containing conditioned media (secreted IGF-II rich). We then immunoprecipitated EphB4 using an antibody against its intracellular domain, followed by a clearing step with an EphB4 antibody recognizing its extracellular region. We compared MSTO211H lysates from cells cultured in either the absence or presence of an IGF-II neutralizing antibody to specifically test the interacting proteins’ dependence on the IGF-II autocrine signal; see the workflow in Figure 1a. The result of this approach through analysis of proteomic data (Figure 1b,c) provided a set of unique EphB4 interacting proteins with peptide representation spanning from 1 to 29 and protein coverage spanning from 19.61 to 49.39 and responsive to IGF-II autocrine signal inhibition in MSTO211H (Table 1). 

As shown in the heat map in Figure 1b, using the selected experimental design, we found a set of EphB4 intracellular interactors in MSTO211H to increase their binding to EphB4 upon IGF-II signal deprivation, while other interactors had an opposite trend, allowing us to further narrow our search and directing the following confirmatory approach. Panther software v 11 analysis of the dataset is conveyed in Figure 1c. Along with a number of structural and cytoskeletal proteins, this approach identified signaling proteins (7%), enzyme modulators (7%), transferases (4%), ubiquitin/proteasome-related factors (3%), chaperones (3%), transporters (3%), calcium-binding proteins (3%), receptors (2%), membrane trafficking proteins (2%), transfer/carriers (2%), and cell adhesion molecules (0.1%). We then focused on the identified ubiquitin-regulated components associated with EphB4, namely, an E1 ligase (UBA1), an E2 ligase (UBE2N), and the putative E3 ligase (DTX3) along with a chaperon (cdc48/p97), which was due to their potential role as specific mediators of EphB4 protein degradation in response to IGF-II-deprivation. Cdc48/p97 is a selective scaffold/scavenging chaperone and triple ATPase/unfoldase involved in the extraction and degradation of ubiquitinated proteins [31,33,34]. To strengthen its specific involvement with the other ubiquitin-related components identified through mass spectrometry (Table 1), we performed a database search (UniProt KB) [35,36] for previously described cdc48/p97 interactions which could provide a mechanistic hypothesis for EphB4 protein rapid degradation upon IGF-II signal deprivation. The UniProt database search for proteins with potential involvement in the observed mechanism confirmed Cdc48/p97 (known as Valosin-containing protein, or VCP) as a specific interactor, as it was previously found to interact independently with both UBE2N and DTX3L, a DTX3-related gene. 

### 2.2. Discovery of a DTX3-UBE2N-UBA1-Cdc48/p97 Protein Complex Docking to the IGF-II Signal-Deprived EphB4 Phospho-Degron/PDZ C-Terminal Domain 

In order to further validate and strengthen the proteomic findings and based upon our previous demonstration of a consensual higher ubiquitination level associated to the rapid EphB4 degradation in cancer cells [26], we studied the effect of the autocrine IGF-II signal on the specific recruitment of the proteomic-identified factors using a synthetically generated and tagged EphB4 c-tail. This EphB4 region was obtained by peptide synthesis and was covalently bound to biotin on its N-terminal side to allow Streptavidin beads pull down (see methods). The use of a synthetic EphB4 tagged peptide as bait in order to study the IGF-II regulated recruitment of the ubiquitin-related factors was chosen in order to overcome the stoichiometric limitation of EphB4 rapid degradation observed with its native protein upon IGF-II blockade. Therefore, we focused on the proteins represented in the proteomic dataset involved in the EphB4 ubiquitin-regulated degradation process (Table 1). In order to maximize both phosphorylative and ubiquitinylation events responsible for the ex vivo degradation complex formation, we designed a novel assay which combined both in vivo and in vitro optimized conditions. To this end, we combined the synthesized biotin-tagged EphB4 C-Degron/PDZ-binding domain as substrate/bait in a cell-free reaction against in vivo treated cell extracts. We further included steps, conditions, and reagents in the in vitro workflow to enhance native enzymatic priming by both endogenous ubiquitin ligases as well as by phospho-Tyr/Thr/Ser kinases present in the MSTO211H extracts obtained from cells upon conditional IGF-II signal block treatment. In this assay, the extracts served both as a source of enzymatic activities (kinases, Ub-ligases, etc.) as well as a source of endogenous complex proteins interacting with the EphB4 c-tail spanning EphB4 region 958–987. This approach (PTM-enhanced protein pull-down) allowed us to study the native components involved, similarly to a co-immunoprecipitation assay, except without the limitations linked to the binding of native proteins to regions of the target protein not of interest (e.g., in our case, the EphB4 extracellular domain). Using this strategy (see the assay workflow in Figure 2a and method description), we tested the IGF-II dependence of UBA1, UBE2N, UBE2L3, DTX3, and Cdc48/p97 binding to the EphB4 C-Degron/PDZ-domain in MSTO211H cells. The results of this substrate/bait study are summarized in Figure 2b. Specifically, it was found that upon autocrine IGF-II signal deprivation, UBA1, DTX3, UBE2N, and Cdc48/p97 binding to EphB4′s 30 C-terminus amino acids which, we have previously found to bear a UBX-like motif contiguous to its PDZ-binding tail [26], was markedly increased. Interestingly, in the case of DTX3 a laddering effect was observed upon IGF-II block on the PTM-enhanced pull down, which was likely due to multiple post-translational modification in its protein occurring in response to IGF-II signal deprivation. The specific involvement of the putative E3 ligase DTX3 in the EphB4 degradation complex was further confirmed by bait-capture ELISA and by co-immunoprecipitation, as shown in Figure 3. This approach confirmed the association of DTX3 with the individually isolated complex components as well as the dependency of such binding from the autocrine IGF-II stimulatory status (Figure 3a–c).

### 2.3. Cdc48/p97 Recruitment to EphB4 in MSTO211H Depends on an Intact Autocrine IGF-II Signal and Requires Cdc48/p97 Triple ATPase Activity

We used co-immunoprecipitation in the presence of EphB4 non-limiting target protein amounts (see methods) in order to confirm dependency of the IGF-II signal deprivation with respect to its interaction with cdc48/p97 in MSTO211H (pre-treated, with a proteasome inhibitor (MG132) and a selective cdc48/p97 triple ATPase/unfoldase inhibitor (ML240), respectively). For this approach, we used cross-linked whole cell extracts from MSTO211H, either pre-treated or not with an IGF-II neutralizing antibody, before harvesting (as specified in methods). As shown in Figure 4, the selective inhibition of triple ATPase Cdc48/p97, which has previously been shown to be critical for its target proteins unfolding [37] but does not generally block the UPS using MG132 [38], was able to interfere with EphB4 recruitment upon IGF-II signal deprivation.

### 2.4. MSTO211H and HelaS3 Express Different DTX3 Isoforms and MSTO211H Expresses a Novel DTX3 E3 Ligase N-Terminal Variant

Considering the different response to the autocrine IGFII block observed between MSTO211H and HelaS3 [26] and due to the fact that DTX3 is known to exist in two isoforms (known as DTX3a and DTX3b) differing in a short aa stretch at the N-terminus end (corresponding to 10 aa from exon 1 in isoform b replaced by 7 aa of the alternatively spliced exon 2 in isoform (a), we decided to further look at this E3 ligase expression in both MSTO211H and HelaS3 as a potential molecular discriminant for the previously observed differential EphB4 degradation behavior between these two cell lines. Specifically, we, hypothesized that different isoform expression between MSTO211H and HelaS3 could underlie the different responsivity to IGF-II autocrine stimuli. To test this hypothesis, we designed DTX3 isoform-specific primers at the level of the predicted coding sequence near the N-terminus end for each of the two DTX3 known isoforms (DTX3a and b) and performed parallel retro-transcription starting from total RNA isolated in MSTO211H and HelaS3 cells, respectively. This was followed by PCR amplification of the resulting cDNA using the same isoform-specific forward primers for either the a or b isoform paired with a common reverse primer spanning a shared gene region at about 300bp within the DTX3 coding sequence (Figure 5). To confirm the specificity of these DTX3 isoform specific primers, we tested them on a DTX3a plasmidic gene clone obtained from Harvard Medical School (clone HsCD00422152). As expected for the control plasmid, the DTX3a amplicon was observed only in the reaction with the DTX3a isoform-specific primers (Figure 5b), while no amplicon was produced when the DTX3b-forward primer was used. The results of this strategy on MSTO211H and HelaS3 are summarized in Figure 5. While HelaS3 clearly was shown to express a DTX3a isoform, interestingly, MSTO211H displayed both the alternatively spliced exons, raising the possibility of the underlying nature of the transcript possibly being consistent with co-expression of both isoforms in MSTO211H. In order to further confirm the identity of the DTX3 isoform sequence, we used Sanger sequencing of the PCR product (as visualized in the gel) in MSTO211H. The analysis of our sequenced transcript using the ExPASy Translate tool [39] unexpectedly revealed an amino acid sequence containing the first nine N-terminal amino acids of DTX3b with a deletion of the corresponding Lysine in position 10, immediately followed by seven amino acids corresponding to the N-terminal portion of DTX3a contiguous and in frame with the remaining part of the DTX3 protein sequence common to both isoforms, suggesting possible exon retention. Further analysis of our cDNA variant against the known human DTX3 gene (ENSG00000178498) and corresponding transcript (ENST00000337737.7) from Ensembl (European Bioinformatics Institute (EMBL-EBI) revealed a g>a mis-sense (transition) mutation corresponding to the DTX3 genomic position on chromosome 12 (NC_000012.42, genomic position 12:57606864) and position 29 of our cDNA variant. We next determined whether this transition from g>a could result in an alteration in splicing, for which we used the splice site prediction program Human Splicing Finder v3.1 (INSERM, 2018) [40]. Mutational analysis of the known human DTX3 gene from Ensembl in the region surrounding the N terminus revealed alteration of an exonic ESE site associated with sequence mutation of the corresponding splice site at the g>a transition in our sequence and at the corresponding position in the reference. Analysis of the COSMIC database revealed no known variants at genomic position 12:57606864, nor between 12:57606861 and 12:57606867. The g>a transition apparently resulted in the observed deletion of lysine (K) in position 10 of DTX3b and the retention of intron 2 (DTX3a), supporting the identification of a new variant. The result of this analysis (summarized in Figure 5) confirmed that the atypical DTX3b primer-amplified variant cloned in MSTO211H was indeed a new N-Terminal DTX3 isoform, which we called DTX3c and submitted accordingly to the NCBI GenBank repository. A reference number has been assigned to the newly submitted partial sequence (BankIt2126934 DTX3C MH536518), and accession details can be found in the NCBI public database [41]. A sequence comparison between the three isoforms is provided graphically and through NCBI BLAST multiple alignment in Figure 5d. In order to confirm the full sequence nature of the newly discovered N-terminal variant, we obtained the full DTX3 transcript’s cds upon PCR amplification and further submitted the complete sequence to GenBank, which assigned an updated reference ID number (MH536518.2) (Figure 6).

### 2.5. Three-Dimensional Modelling of E3 Ligase DTX3 N-Terminus Domain Variants Disclose Different Preferential Folding for DTX3c

In order to shed light on the nature of DTX3c with respect to the observed differences in DTX3 recruitment by EphB4 between MSTO211H and HelaS3, we further compared DTX3 isoforms through a structure prediction approach using the ROSETTA algorithm (refer to [42] and methods) in order to obtain and compare the N-Terminus three-dimensional structures of the DTX3 isoforms at an atomic level of detail. To this end, we first predicted the intrinsically disordered regions using the PrDOS web server [43] [http://prdos.hgc.jp accessed on 18 October 2018]. The results (Figure 7a) show that the first 100 residues in the chain likely possess secondary and tertiary structures (disorder probability lower than 50%) followed by a proline-rich disordered region of 57 residues. The structure of the remaining C-terminus domain has been already determined using X-Ray diffraction (PDB structure 3PG6 [https://doi.org/10.1002/prot.24054 accessed on 2 October 2018]). Thus, we focused on the characterization of the N-terminus domain for the three different isoforms. Because ROSETTA is based on stochastic optimization of its machine learning prediction model, the most reliable results are obtained when a large number of solutions are generated rather than a single one; therefore, we calculated the structure of 40,000 theoretical models for each isoform and post-processed the dataset using clustering to identify the most representative configurations (see methods for details). Visual renderings of the most representative structures obtained through this process are shown in Figure 7c–f). The graph in Figure 7b summarizes the results of the dynamic modeling, with conformations shown as circles displaying the clustered representation of the major probable conformations and the distance between any two nodes of the graph being inversely proportional to their structural similarity. The numbers assigned to each conformation in Figure 7f were used to rank the conformations in terms of the occupancy of the parent cluster conveyed in the graph in Figure 7b (i.e., the lower the number, the larger the size of the cluster). Interestingly, the most populated clusters of isoforms DTX3a and DTX3b (models A1 and B1, pairwise RMSD = 5.5 Å) display a very similar fold, and both show similar significant distance from isoform DTX3c (RMSD = 12.2 Å between A1 and C1 and RMSD = 12.9 Å between B1 and C1). This is evident from the 3D rendering in Figure 7c–e (with the DTX3c N-terminus domain highlighted by the red dotted circle). This observation suggests that the isoform DTX3c N-terminus domain is likely to fold into a different 3D structure in vivo compared to isoforms DTX3a and DTX3b. These results further support our working hypothesis that DTX3c could be the molecular discriminant underlying the differential EphB4 recruitment previously observed between MSTO211H (DTX3c-expressing cells) versus HelaS3 (DTX3a-expressing cells).

## 3. Discussion

Our observation of an autocrine IGF-II-IR^A^-specific signal targeting the EphB4 C-terminal phospho-inhibited Degron bearing a UBX-like motif immediately upstream of the PDZ-binding motif that allows sustained ectopic expression of EphB4 in mesothelioma cancer cells [26] is in line with the present findings identifying a defined set of ubiquitin E-ligases recruited at the same EphB4 region upon autocrine IGF-II signal deprivation. The present study adds a great deal of novel molecular detail to this ongoing scenario, starting with the identification of a novel isoform of DTX3 E3 ligase (DTX3c) and its direct involvement with the IGF-II–EphB4 signaling axis in MSTO211H cells. A graphical summary of this scenario is conveyed in Figure 8. The importance of E3 ligases in regulating the expression of a number of cellular proteins has been previously established [44,45], especially in pathological contexts such as cancer [46,47,48]; therefore, the specific identification of E3 ligases and their exact targeted degrons among Ub-regulated cellular proteins has emerged as a leading objective in the cell biology field [49,50]. Until recently, DTX3 E3 ligase activity had been considered putative, as it had not yet been associated with the degradation of a specific protein. Recently, however, it has been found to physically interact with Notch2 by Yeast-Two-Hybrid assay and to inhibit Notch2-mediated proliferation in esophageal carcinoma [13]. Therefore, to the best of our knowledge and based on the literature search, the present findings provide novel evidence of the direct involvement of the DTX3 E3 ligase in a cellular pathway other than the previously described association with Notch signaling [8,9,10,11,12,13]. The present study, along with the data provided in Scalia et al. 2019 [26], physically and functionally associate DTX3 to the observed degradation of EphB4 under the control of a different ligand/receptor system, namely, the autocrine IGF-II/IR^A^ system. Specifically, we demonstrate that recruitment of the novel DTX3c-E3 ligase to the EphB4 C-terminal degron/PDZ-binding region in MSTO211H cells, which display sole expression of this new isoform, is highly dependent upon IGF-II signal neutralization and correlates to the previously observed increases in EphB4 c-tail ubiquitination and overall protein degradation [26]. The present findings are in line with the hypothesis that DTX3c might constitute a molecular discriminant towards the observed EphB4 protein stability and expression level dependence on the IGF-II signal in MSTO211H, and not in HelaS3. Further investigation is needed to confirm this relationship in cancer, for example through targeted proteomic screening in patient-derived cells. On the mechanistic side, it has been widely established that degradation of a specific protein via the ubiquitin/proteasome pathway requires the sequential and/or reciprocal action of an E1, E2, and E3 ligase [3,4]. This further strengthens the present finding of UBA1 (E1) and UBE2N (E2) along with the new DTX3 variant (E3), having been established in the present study as EphB4 C-Degron direct binders under autocrine IGF-II signal-deprived conditions. We have further identified Cdc48/p97 ATPase/unfoldase as an intrinsic component of this complex binding to the EphB4 C-Degron/PDZ-binding motif region under the same growth-factor regulatory conditions. Indeed, Cdc48/p97 has already been shown to trigger selective unfolding and extraction of a number of target cellular proteins towards their UPS-mediated degradation [51,52,53]. Our present data support Cdc48/p97 involvement in EphB4 protein levels regulation and show the dependence of its ATPase enzymatic activity in EphB4 complex recruitment towards potential tertiary structure modification(s). Interestingly, the ATPase-dependent recruitment pattern of Cdc48/p97 to the EphB4 C-Degron shown herein parallels the previously observed rapid IGF-II deprivation-mediated EphB4 proteolysis [26] which we found to be associated with dephosphorylation of its C-terminal tail residue (Tyrosine 987) overlapping the previously described “PQY” PDZ-binding motif [27]. Overall, the identification of the EphB4 phospho-inhibited degron [26] and its C-tail-interacting ubiquitin E-Ligase complex provides a potential mechanism for the modulation of EphB4 pro-angiogenic and pro-invasive effects. Indeed, a preliminary analysis of the published studies providing evidences of in vivo modified PTM residues in large proteomic studies, allowing us to pinpoint the most common PTM modifications to the level of the last 30 aa of EphB4 (unpublished data). This analysis suggests a potential role for Lysine 973 and Threonine 976 along with Tyrosine 987 in the C-terminal PDZ-binding motif in the observed IGF-II-regulated circuitry. Furthermore, the partial proximal overlap between the EphB4 C-Degron and the previously described SAM domain suggests a functional link between EphB4 levels and protein dimerization according to the IGF-II/IR^A^ signal status. Further investigation will eventually clarify the dynamic interactions and molecular events linking IGFII/IR^A^ upstream signaling to EphB4 and the contribution to its biological effects in the cancer cell. As we previously demonstrated that IGF-II triggers Tyr987 dephosphorylation via the IR^A^ signal and not via IGFR signal using genetically modified cellular models [26], it will be relevant to shed light on the differential effects of the IGF-II/IR^A^-mediated signal on EphB4 as compared to that mediated by the IGF-II/IGF1R, as well as the role of related receptorial systems such as the IR/IGF1R hybrids using the same ligands and signaling adaptors [54]. A tentative hypothesis supported by our findings is that the IGF-II/IR^A^ signal may play a specific role in EphB4 steady-state protein expression levels, while the IGF-II/IGF1R may have regulatory effects on EphB4 kinase-mediated signals (e.g., cell adhesion/migration). According to the present scenario, and requiring further investigation, the IGF-II/IR^A^ autocrine loop could provide a post-translational pathway aimed at gaining and maintaining high ectopic levels of the EphB4 protein in a number of cancer cells co-expressing such autocrine loop circuitry and degradation factor components [26,55]. Ultimately, our findings offer the following: (a) identification of a novel cancer-associated DTX3 isoform (DTX3c) by disclosing a new biological/pathological context for the DTX3-E3-ligase and (b) demonstration of the involvement of UBA1, UBE2N, and Cdc48/p97 as part of the identified complex conditionally docking to the EphB4 C-terminal Degron/PDZ-binding region upon inhibition of the IGF-II-IR^A^ autocrine cancer loop. Future elucidation of the specific role of this newly described ubiquitin–ligase complex towards controlling EphB4 protein levels may provide novel molecular agents and targeting strategies towards down-modulation of the pathological effects associated with EphB4 in cancer and human disease.

## 4. Materials and Methods

### 4.1. Tagged Peptide Synthesis

N-Biotin EphB4 958–987 was obtained from New England Peptides Inc. (Gardner, MA, USA). The Biotin-conjugated peptide sequence corresponding to the last 30aa of EphB4 C-tail was Biotin-NH-QKKILASVQHMKSQAKPGTPGGTGGPAPQY.

### 4.2. Cell treatments and Exogenous Recombinant/Tagged Proteins Expression

Hela S3 was obtained from ATCC (Manassas, VA, USA); MSTO211H cells were obtained from Fox Chase Cancer Center and underwent intramural confirmatory STR genotypization. Cancer cell lines were maintained in RPMI, 10% Fetal Bovine Serum (SIGMA, St. Louis, MO, USA). Secreted IGF-II conditioned media were obtained by collecting the culture media of the selected IGF-II secreting cancer cell lines at a minimum of 85% confluency. Autocrine IGF-II loop neutralization was obtained by adding a neutralizing anti-human IGF-II antibody (clone AF292, R&D systems, Minneapolis, MN, USA) to the conditioned culture media of IGF-II-secreting cancer cells, and the neutralization effect was evaluated following 12 h treatment. Following treatment, the CM media were collected and stored at −20 °C. Cells were washed with cold PBS and harvested by adding ice-cold lysis buffer (see below), collected by scraping, and lysates cleared by full-speed microfuge centrifugation at 4 °C. Protein amounts were measured using an aliquot of the cleared lysates using BCA reagent (SIGMA, Saint Louis, MO, USA), and lysates used for protein experiments were brought to the same the concentration using ice-cold lysis buffer, with full protease/phosphatase inhibitor cocktail used throughout the study (see complete recipe below). Total cell extracts used in this study were obtained with NP40 containing lysis buffer (1% NP40, NaCl 137 mM, 20 mM Tris 7.4, 1 mM MgCl, 1 mM CaCl_2_, 10% Glycerol) with protease/phosphatase inhibitors (1 mM DTT, 2 mM Benzamide, 10 mM b-glycerol Phosphate, 10 mM NaF, 2 mM NaVO_3_, 10 μg/mL Pepstatin, 10 μg/mL Leupeptin, 10 μg/mL Aprotinin, 100 nM Okadaic Acid, 0.5 mM PMSF). MG132 and ML240 inhibitors were dissolved in appropriate solvents as per the manufacturers and diluted in culture media at a final concentration of 200 nM, which is described in the literature as efficiently inhibiting both in vivo and in vitro enzymatic activities [38,56].

### 4.3. Proteomic Analysis of EphB4 Immuno-Complex

Label-free modified in-stage tip (iST) proteomics studies were performed using the proteins recovered by immunoprecipitation from MSTO211H total cell lysates upon reversible cross-linking using anti-EphB4 antibody-coated prot A/G beads, following the workflow shown in Figure 1. Briefly, 6 M guanidinium hydrochloride buffer was added to the lysates, which were than heated for 5 min at 95 °C. Lys-C was added. The lysate was diluted five-fold with 25 mM Tris, 10% acetonitrile solution. The proteins were digested for 4 h at 37 °C and digestion was achieved by overnight incubation at 37 °C with trypsin [57]. Digestion was stopped by acidification with 3% TFA (trifluoro acetic acid). The incubated solution was centrifugated at 2000× *g* for 5 min. The supernatant consisting of peptides was loaded onto an activated cation stage tip made in-house. The peptides were eluted into six fractions [57,58]. The de-salted tryptic peptide samples were loaded onto an a QExactive Thermo Scientific Mass Spectrometer with Acclaim PepMap 100 pre-column (C18, 75 μm × 2 cm, Thermo Scientific Waltham, MA, USA) and separated using an Easy-Spray PepMap RSLC C18 column (2 μm, 50 μm × 150 mm, Thermo Scientific) with an SS emitter via Easy nLC system with Easy Spray Source (Thermo Scientific). To elute the peptides, a mobile-phase gradient was run using increasing concentrations of acetonitrile. Peptides were loaded in buffer A (0.1% (*v*/*v*) formic acid) and eluted with a nonlinear 145-min gradient as follows: 0–25% buffer B (0.1% (*v*/*v*) formic acid, 85% (*v*/*v*) acetonitrile) for 80 min, 25–40% B for 20 min, 40–60% B for 20 min, and 60–100% B for 10 min. The column was then washed with 100% buffer B for 5 min, 50% buffer B for 5 min, and re-equilibrated with buffer A for 4 min. The flow rate was maintained at 300 nl/min. Electron spray ionization was delivered at a spray voltage of −2000 V. Ms/Ms fragmentation was performed on the five most abundant ions in each spectrum using collision-induced dissociation with dynamic exclusion (excluded for 10.0 s after one spectrum), with automatic switching between Ms and Ms/Ms modes. The complete system was entirely controlled by Xcalibur software version 4.1 (Thermo Fisher, Waltham, MA, USA, accessed on 19 July 2020). Three purely qualitative technical replicates were generated (no quantitative analysis was needed). Mass spectra processing was performed using Proteome Discoverer v2.4, accessed on 19 July 2020) The generated de-isotoped peak list was submitted to two separate databases, namely, the Mascot (Mascot Stable release: 2.6.00/December 2016) and Sequest HT (2013 version, University of Washington, Seattle, WA, USA) databases. For both databases, the search parameters were set as follows: species, homo sapiens; enzyme, trypsin with a maximum of two missed cleavages; minimum peptide length, 4; maximum peptide length, 144; static modification, carbamidomethyl/+57.021 Da(C); 10 ppm precursor mass tolerance and 0.6 Da fragment mass tolerance for Ms/Ms fragment ions. Quantified proteins were selected and clustered by biological functions using Ingenuity Pathway Analysis software version 2.2.1 (Qiagen, Germantown, MD, USA, www.ingenuity.com, accessed on 19 July 2020) for bio-informatics analysis.

### 4.4. Reversible Cross-Linking of Native Cell Extracts

Reversible cross-linking was performed following prot A/G preclearing at 4 °C in the presence of SPDP (*N*-Succinimidyl 3-(2-pyridyldithio)propionate) for 2 h in gentle rotation. The stop and reversion of the covalent native cellular proteins cross-linking was performed following an immunoprecipitation step with the primary antibody-prot A/G beads by adding 20 mM Tris, pH 7.5 (final concentration) followed by ice incubation for 10 min.

### 4.5. Covalent Cross-Linking of Primary Antibodies and Protein Co-Immunoprecipitation

Primary antibodies used for immunoprecipitation were covalently and irreversibly linked to solid support by combining 0.2 mg of purified monoclonal antibody with 100 μL of prot A/G slurry (dry volume) and equilibrating in 150 mM Borate, pH 9.0, for 1 h at room temperature (RT) by gentle rotation. Irreversible cross-linking was carried out with 20 mM Dimethyl pimelimidate in Borate buffer, pH 9.0, for 30 min at RT followed by incubation with 0.2 M ethanolamine, pH 8.0 for 1 h at RT. The newly generated immune-beads were resuspended and stored at 4 °C in PBS containing 0.01% sodium azide for mid-term to long-term storage. Working aliquots of the resulting Ab-Prot A/G beads were resuspended in immunoprecipitation buffer before use. In order to evaluate the cross-linking efficiency, 5 μL aliquots of the pre- and post-cross-linking suspensions were separately denatured at 95 °C in gel loading buffer containing 50 mM DTT and used for SDS-PAGE, followed by Comassie staining of the resulting gel as previously described in [59]. Immunoprecipitation of native complexed proteins was performed by adding 10 μL of covalently linked primary antibody-prot A/G beads to the cell extracts following protein A/G preclearing, followed by gentle rotation O/N at 4 °C. Immuno-complexed beads were collected by low-speed centrifugation and resuspension of the beads in SDS-gel loading buffer containing 50 mM DTT. Complexed proteins were released from the immuno-beads by denaturation at 94 °C for 5 min, brief vortexing, and full speed centrifugation. The supernatant was loaded onto polyacrylamide gel and proteins were resolved by electrophoresis followed by western blotting, as described below. Beads alone with no primary Ab was used as a negative control.

### 4.6. Western Blotting of Co-Immunoprecipitated/Pulled-Down Proteins

Western blotting was performed as previously described [60] with post-protein transfer onto PVDF membranes (Hibond, GE-Amersham, Pittsburgh, PA, USA). Non-specific blocking was performed with 5% Bovine Serum Albumin in PBST. Following the Albumin blocking step PVDF-transferred proteins with clearly distinct molecular weight were tested on sub strips of the same membrane to minimize inter-experimental bias and maximize post-IP sample utilization. To this end, membrane strips generated by scissor cutting at indicated molecular weight markers were tested with designated primary antibodies (Appendix A) at the specified concentration/dilution range from 1:1000 to 1:2000 in PBST. HRP-conjugated antibodies (species-specific, Rockland, Pottstown, PA, USA) were used at 1:2000 dilution. The chemiluminescence signal was generated with “Supersignal West Fempto” substrate (Thermo-Scientific, Waltham, MA USA) and detected using an Odyssey Fc digital gel detection system (Li-Cor Bioscience, Lincoln, NE, USA).

### 4.7. EphB4 C-Degron Bait/Interaction Study via Novel Post-Transcriptional Modification-Enhanced Pull-Down Assay

Proteins specifically interacting with the EphB4 958–987 C-terminal Degron were studied by using an N-Biotin conjugated synthetic peptide (New England Peptides, Gardner, MA, USA) reflecting the native corresponding sequence of human EphB4 (958-QKKILASVQHMKSQAKPGTPGGTGGPAPQY-987). This was used as bait in a novel modified pull-down/ubiquitination/phosphorylation in vitro assay designed to enhance the priming post-transcriptional modifications (tyrosine ubiquitination and phosphorylation) induced by the cancer cell extracts obtained from the autocrine IGF-II-deprived cancer cells versus PTM induced by cell extracts obtained from parallel untreated cells (with intact autocrine IGF-II stimulus). For the pull-down assay, 50 μg of synthetic peptides were first coupled with 6.25 μL of Streptavidin magnetic beads (Genscript, Piscataway, NJ, USA) for 2 h at 4 °C rotation followed by a PBS wash. The STP beads-coupled N-Biotin EphB4 958–987 bait was then used to assemble a 100 μL mixture containing 750 μg whole cancer cell extract (enzymatically active), 10 mM Tris pH 7.5, 4 mM ATP, 4 mM ATP, 2 mM MnCl_2_, 1 μg human recombinant Ubiquitin (Boston Biochemical, Cambridge, MA, USA), and 0.025% NP40 along with a complete protease/phosphatase inhibitor cocktail. Protein–protein interactions along with cell extract-induced tyrosine phosphorylation/ubiquitination post-transcriptional modifications were allowed to occur by incubating the mixture for 48 h at 4 °C in gentle rotation. Beads with no cell extract were used as a negative control for the assay. At the end of the incubation, the protein-bound beads were washed once with PBS 0.05% NP40 with protease/phosphatase inhibitors followed by denaturation in SDS gel loading buffer containing 50 mM DTT at 94 °C for 3 min. The denatured product was fully released from the beads by brief vortexing followed by low-speed centrifugation and loaded onto 12% denaturing polyacrylamide gel (Thermo, Waltham, MA, USA). The resolved products were transferred onto PVDF. PDVF membrane and cut into squares corresponding to the target protein according to their molecular weight following an Albumin non-specific binding blocking step (5% Albumin in PBST), and proteins were detected by western blotting using specific antibodies and conditions, as listed in this section.

### 4.8. Protein Complex Interaction Studies by ELISA/EIA

Selected protein–protein interactions detected by proteomic analysis, co-immunoprecipitation, and pull-down assay were confirmed by ELISA [55]. Immune-capture antibodies used for coating (0.3 μg/mL) were adsorbed onto 96-well plates (Maxisorb™, Nunc Millipore Sigma) in H_2_CO_3_/HCO_3_- buffer, pH 9.6 o/n at 4 °C followed by non-specific site blocking in 20 mM Tris pH 7.4, 150 mM NaCl, 0.05% Tween20, 1% Bovine Serum Albumin at 56 °C for 30 min. In the presence of protease/phosphatase inhibitors, 10–40 μg cell lysates were incubated for 2 h at 4 °C Detection of associated proteins was pursued with primary antibodies at 1:500 dilution in 50 mM Hepes pH 7.5, 150 mM NaCl, 0.05% Tween20, 1% Bovine Serum Albumin with protease/phosphatase inhibitors incubating for 2 h at 22 °C. Species-specific secondary antibodies were used at dilution, and were conjugated with HRP diluted in the same buffer used for primary Ab incubation. The ELAST Elisa Amplification System (Perkin Elmer, Waltham, MA, USA) was used depending upon the resulting signal intensity. Either OPD (Millipore Sigma, St. Louis, MO, USA) or TMB (Thermo Scientific, Waltham, MA, USA) were used with the ELAST Elisa Amplification System (Perkin Elmer, Waltham, MA, USA) as colorimetric substrates for ELISA, and reactions were ended by adding 2M sulphuric acid. Optic Density was quantified by spectrophotometric reading at 492 (OPD) or 450 (TMB) wavelengths in a BIOTEK (Winooski, VT, USA) reader (μQuant). No primary Ab condition was used for negative control for the ELISA. EIA using EphB4 958–987 synthetic peptide as plate-coated bait was adopted to confirm binding of selected proteins from cell extracts to the last thirty amino acid residues in parallel with ELISA using the same associated target protein-directed antibodies (secondary Ab) used for ELISA. Synthetic peptides (2 μg) were used to coat each well plate for the ELISA protocol above in the place of the capture antibody. A no-cell-extract condition was used for determination of non-specific background on EIA and subtracted from the extract-containing conditions. All the following steps were identical to the ELISA protocol up to the detection level.

### 4.9. DTX3 Isoforms cDNA Amplification, N-Terminal Cloning of Expressed Variants, and DTX3 Cds Sequencing

DTX3 isoform expression and partial cloning was carried out in MSTO211H and HelaS3 cell lines. Total RNA was obtained from freshly cultured cell lines using TRIZOL™ reagent (Thermo-Scientific, Waltham, MA, USA) and Direct-zol™ RNA Miniprep Plus column purification kit (Zymo Research, Irvine, CA, USA). cDNA was obtained using the ReadyScript^®^ cDNA Synthesis Mix (Millipore Sigma, Burlington, MA, USA) according to the manufacturer’s instructions on an ABI thermocycler and PCR performed using Accuprime Taq Polymerase (Thermo-Scientific, Waltham, MA, USA) and optimized buffer at 94 °C denaturation (30 s), 55 °C annealing (30 s) and 72 °C extension (45 s) sequential thermocycle for 35 cycles. Total RNA extraction and RT-PCR were carried out under the conditions specified under the homonymous section above. The DTX3a-specific forward primer sequence was 5′ ATG TCG TTC GTC CTG TCC AGA 3′, while the DTX3b specific forward primer sequence (common to DTX3c mRNA) was 5′ ATG CCA ATT CTA AGC TCT TCA GG 3′. A common Reverse primer spanning the region of DTX3a/b was used (5′ CTG AGC TTT CTT CAG CTC TTT C 3′). PCR products were visualized on 1.2% Agarose containing Ethidium Bromide (1:20,000) on a UVB gel detection system. Full cds for DTX3c was obtained using the forward primer common to DTX3b (see above) along with a reverse primer spanning a 3′ region common to all DTX3 isoforms (5′ CAG TCA TCT GTG ATA CCC TTC G 3′). The amplified cDNA was quantified by Nanodrop ND1000 UV/VIS spectrophotometer (Thermo Waltham, MA, USA) and an 800-ng aliquot was used for Sanger Sequencing (GeneWiz, South Plainfield, NJ, USA). The cDNA sequences obtained from Sanger sequencing were translated using the Xpasy Translate bioIT online tool on the SIB Bioinformatics Online Portal (https://web.expasy.org/translate/, accessed on 10 April 2018).

### 4.10. 3D Structure Prediction Study

In order to predict the structure of the N-terminus domain, we used the ROBETTA software program [42], as no experimental structure was available for orthologue genes. We performed ab initio structure prediction each isoform independently. For each of them, a structural fragment library was generated using the ROBETTA webserver [http://robetta.bakerlab.org accessed on 27 September 2018] [61]. The fragments were used afterwards to sample the proteins’ conformational space, generating 4 × 10^4^ independent structures in each case. Of these, we selected the first 2000 structures with the lowest score and clustered them on the basis of their root mean square deviations using the Jarvis–Patrick methods implemented in the GROMACS software package x.1. (https://doi.org/10.1016/j.softx.2015.06.001, accessed on 18 October 2018) [62]. The parameters used for clustering were a cutoff of 1.4 nm and a minimum number of neighbors of 4. These were found to be appropriate and to return a manageable number of clusters. Only clusters with more than 100 structures were finally selected, for a total three clusters for the DTX3a and DTX3c isoforms and four clusters for the DTX3b isoform. The most representative configurations were selected for each cluster.

## Figures and Tables

**Figure 1 ijms-24-07380-f001:**
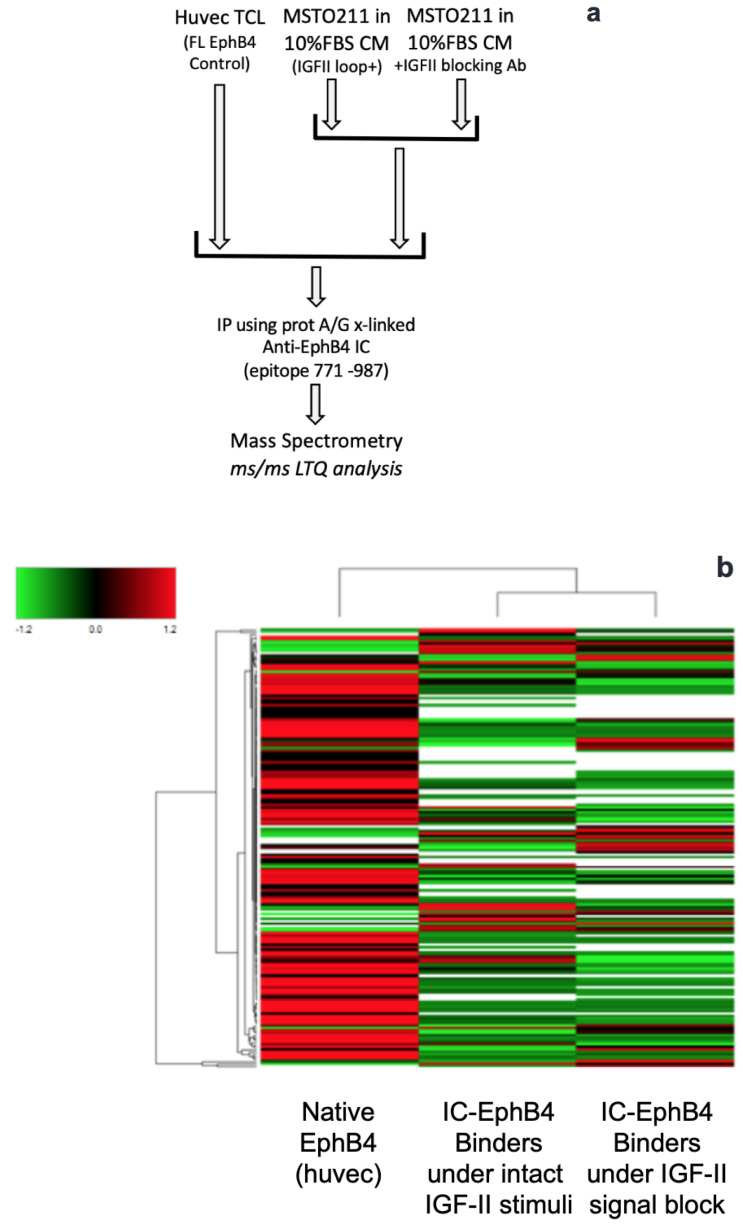
Identification of selective autocrine IGF-II-dependent intracellular EphB4 binders via immuno-affinity proteomics. (**a**) Immune-enriched proteomic experimental workflow; (**b**) heat map of proteins detected by Ms/Ms in EphB4 immune-precipitates upon ex vivo deprivation of the autocrine IGF-II signal; (**c**) Panther^TM^ analysis of the proteins identified by immune-enriched proteomics.

**Figure 2 ijms-24-07380-f002:**
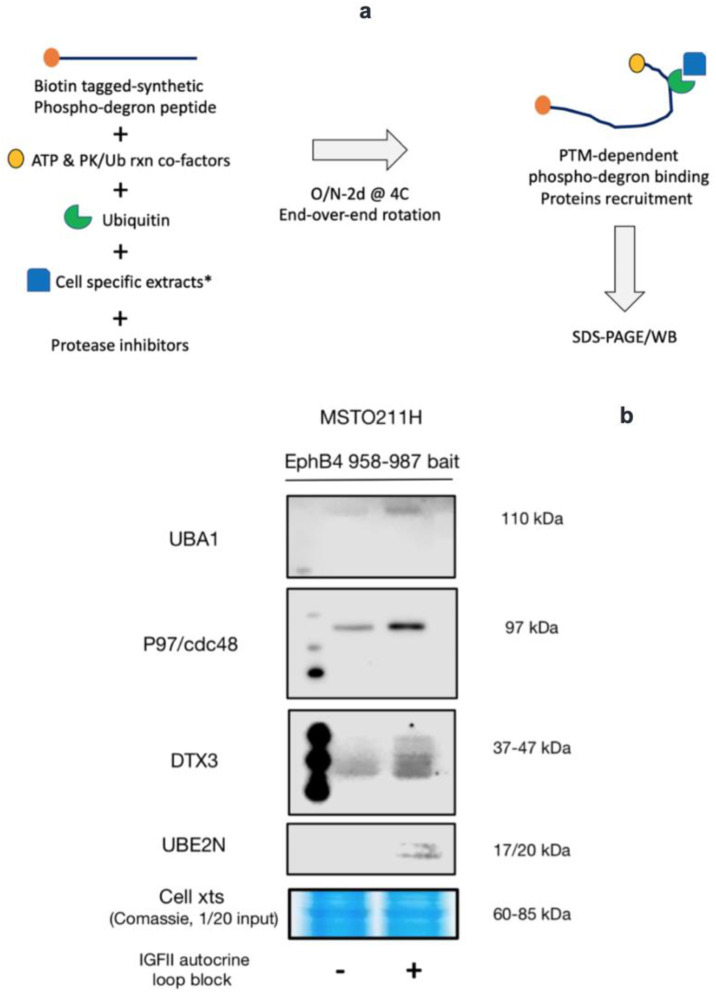
A DTX3-UBE2N-UBA1-Cdc48/p97 protein complex docking to the IGF-II signal-deprived EphB4 Phospho-Degron/PDZ C-terminal region. (**a**) Schematics of the PTM-enhanced EphB4-C-tail pull-down assay. (*) Asterisk in 2a points at the in vivo IGF-II conditional block treatment; (**b**) ability of EphB4 c-Tail to recruit UBA1, UBE2N, DTX3, and p97/cdc48 from MSTO211H lysates following ex vivo autocrine IGF-II loop neutralization.

**Figure 3 ijms-24-07380-f003:**
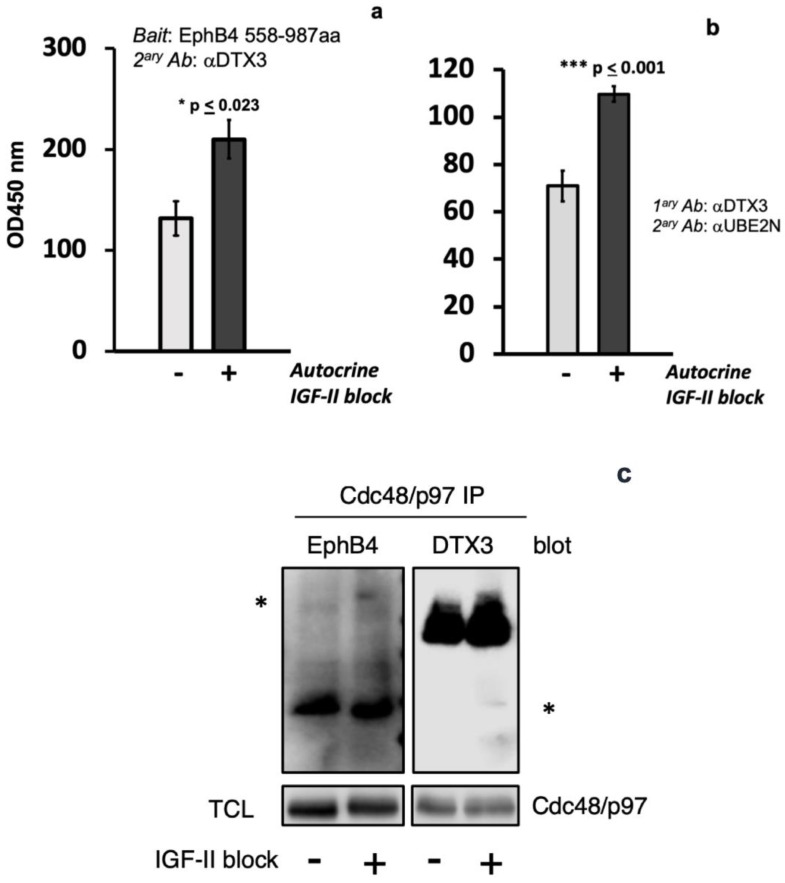
Independent co-purification of DTX3 with EphB4 C-tail, UBE2N, and CDC48/p97 upon autocrine IGF-II block in MSTO211H. Solid phase pull down-EIA (**a**), ELISA (**b**), and immunoblot of co-immunoprecipitated protein (**c**) were used to confirm independent physical interaction of native proteins from MSTO211H lysates. The asterisks indicate the specific expected bands. Quantitation for EIA and ELISA represents mean + s.e.m. of six measurements (*n* = 6). Statistical analyses were performed and *p* values calculated using Student’s *t*-test with unequal variances.

**Figure 4 ijms-24-07380-f004:**
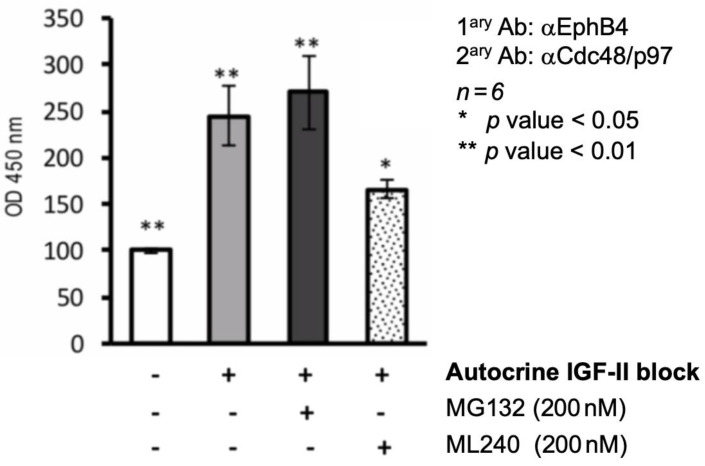
Coprecipitation of EphB4 requires a fully active cdc48/p97 ATPase/unfoldase. An ELISA was set up to capture native EphB4 from pre-treated MSTO211H lysates at a non-limiting target/capturing Ab rate, as described in the methods. This assay was used to detect the level of cdc48/p97 binding to native EphB4 in MSTO211H lysates upon selective inhibition of the proteasome pathway (MG132, 200 nM) or cdc48/p97 enzymatic activity (ML240, 200 nM). The graph shown here is the statistical representation (mean ± s.e.m) of three independent experiments performed in quadruplicate. Statistical analyses and *p* value were calculated using Student’s *t*-test with unequal variances.

**Figure 5 ijms-24-07380-f005:**
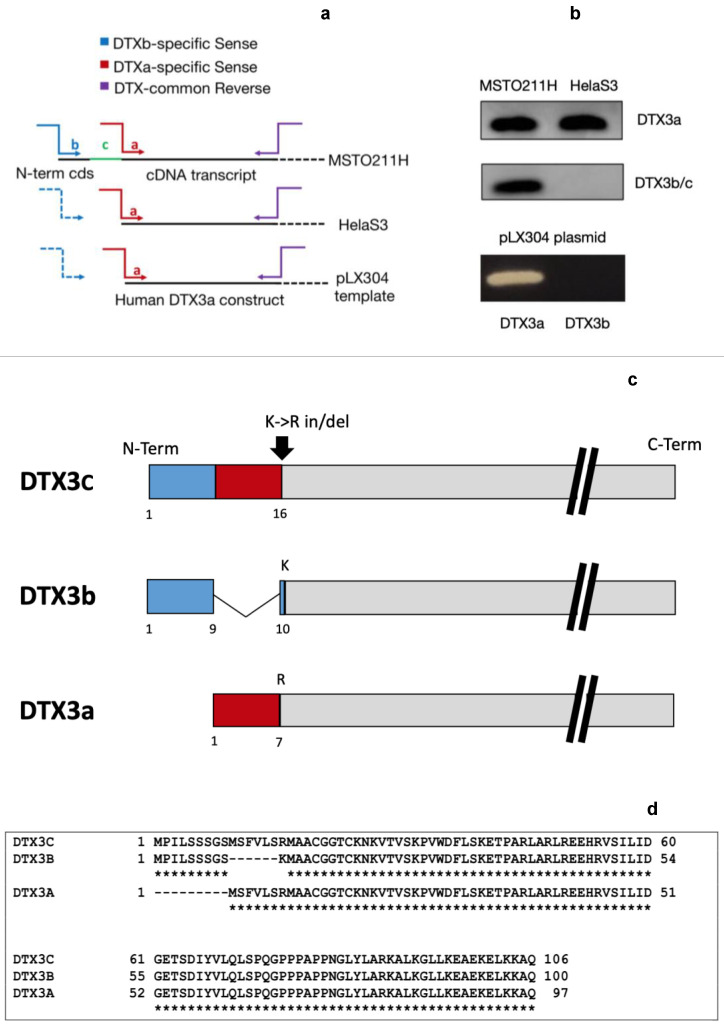
Discovery of a novel DTX3 isoform as the sole expressed variant expressed in MSTO211H. (**a**,**b**) Amplification of DTX3 N-terminal domain with isoform-specific primers uncover the expression of a new variant in MSTO211H. The solid-colored arrows in 5a refer to the isoform-specific primers providing an amplicon due to the presence of the expressed isoform transcript; the dashed arrows relate to the ineffective isoform-specific primer amplification observed in those specimen not expressing the underlying transcript (5b, left panel). The novel transcript (DTX3c) was amplified with both the DTX3a 5’ and the DTX3b 5’ primers due to exon retention of both isoforms N-terminal exons as shown in Figure 2b. (**c**,**d**) Sanger sequencing of the transcript amplified from MSTO211H confirmed the new variant, displaying retention of two alternatively spliced exons and a single point mutation (K>R) on the aa in position 16 due to an in/del event (5c). The asterisks underlying the sequences of the known DTX3 isoforms in 5d emphasize the conserved residues in DTX3c.

**Figure 6 ijms-24-07380-f006:**
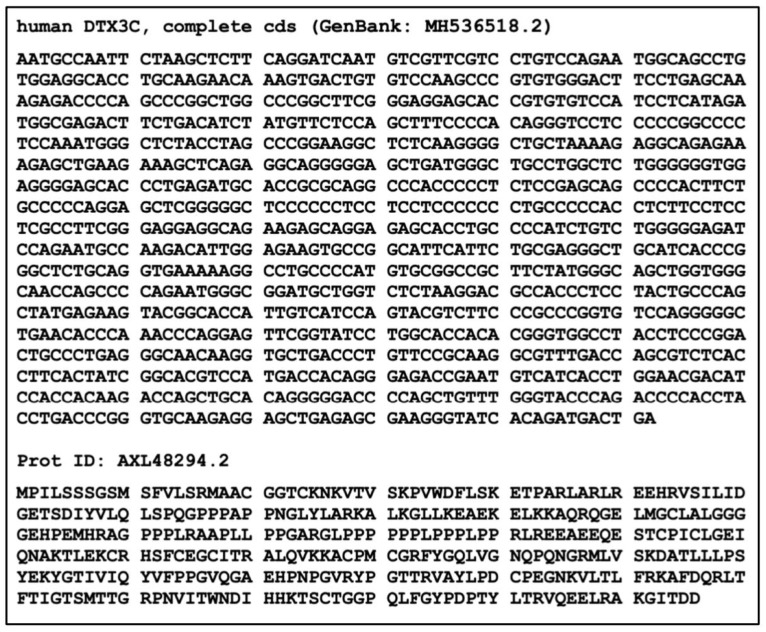
Full sequencing of DTX3C gene and protein translation product. In order to confirm the same sequence of DTX3c to its two isoforms on the remaining central and C-terminal region, a full CDS sequence was amplified in MSTO211H and sequenced as described in the methods. The full sequence and translated product were validated by GenBank upon submission and assigned a ref ID as indicated in the figure.

**Figure 7 ijms-24-07380-f007:**
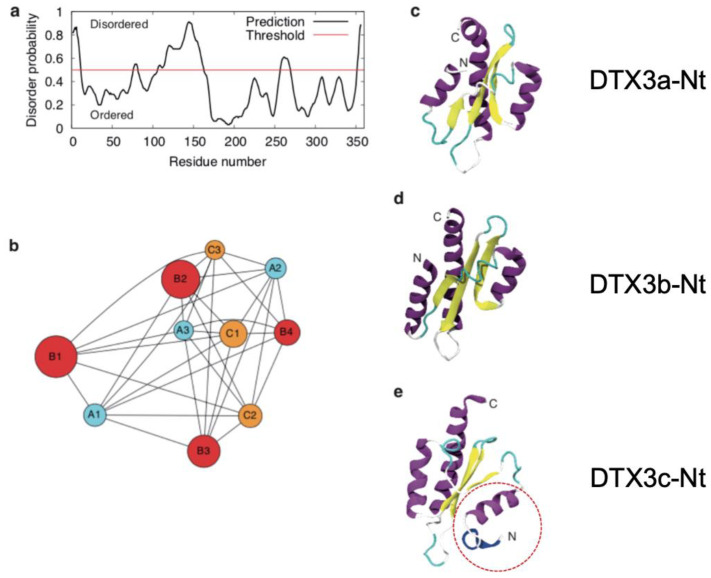
Tridimensional modeling of DTX3c N-terminal domain and comparison with previously identified Isoforms. (**a**) Disorder prediction graph of DTX3c N-terminal region; (**b**) cluster representation for probable conformations and atomic distance of the models generated for the DTX3c N-T region; (**c**) N-Terminal domain rendering for DTX3a; (**d**) DTX3b; (**e**) DTX3c; (**f**) (A1−C3) additional spatial distribution renderings of the three N-terminal domains of DTX3, with (A1−A3) and (B1−B4) displaying close 3D resemblance compared to DTX3c (C1−C3). The results suggest that DTX3c has a unique spatial distribution, implying different biological activities such as EphB4 degradation efficiency. For explanation of red dotted circles, see text.

**Figure 8 ijms-24-07380-f008:**
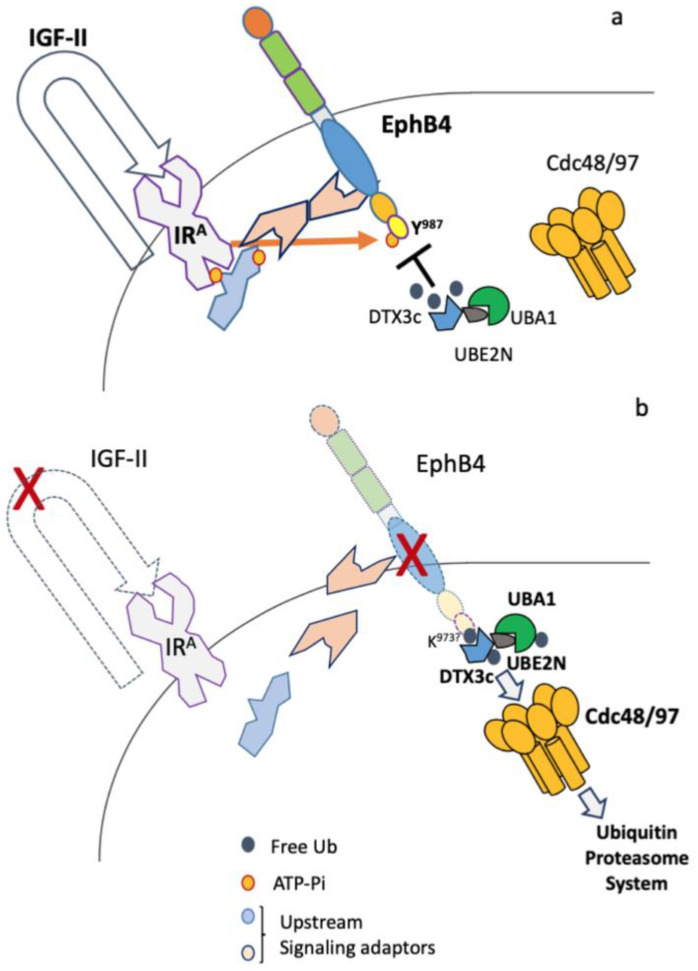
Proposed model for EphB4 over-expression in cancer via IGF-II/IR-A autocrine stimuli. The cumulative scenario deriving from the present work and our previously published work [26] allows us to propose a mechanistic model in which the IGF-II/IR-A autocrine signal is a key condition for EphB4 overexpression in those cells co-expressing the factors described herein. (**a**) Under such self-stimulated conditions (such as those in cancers secreting IGF-II), the IR-A tyrosine kinase can phosphorylate EphB4 on its c-terminal Tyr987 residue. Such phosphorylation has an inhibitory role on EphB4 C-terminal degron recruitment by DTX3c (E3-ligase), causing an extended expression half-life of EphB4 and its cancer-promoting functions. (**b**) Upon IGF-II signal neutralization, EphB4 Tyr987 is rapidly dephosphorylated and its C-tail is ubiquitinylated, allowing DTX3c and its associated degradation partners to exert their UPS-mediated role.

**Table 1 ijms-24-07380-t001:** IGF-II-regulated Ubiquitin-related components identified in EphB4 immune complex in MSTO211H cell lysates by Ms/Ms.

UniProt Accession	Protein	Known Function	MW (kDa)
P22314	**UBA1**	E1 Ub-Ligase	117.8
P55072	**Cdc48/p97**	Unfoldase, protein degradation chaperone	89.32
Q8N9I9	**Deltex3**	E3 Ub-Ligase	38.0
P61088	**UBE2N**	E2 Ub-Ligase	17.1

## Data Availability

New sequences generated in this work were submitted to NCBI GenBank and are publicly available.

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
