# Peer review of "Novel Isoform DTX3c Associates with UBE2N-UBA1 and Cdc48/p97 as Part of the EphB4 Degradation Complex Regulated by the Autocrine IGF-II/IRA Signal in Malignant Mesothelioma"

_ijms, 2023, doi:10.3390/ijms24087380_

Round 1

Reviewer 1 Report

The manuscript titled 'Novel Isoform DTX3c Associates With UBE2N-UBA1 And Cdc48/p97 As Part Of The EphB4 Degradation Complex Regulated By The Autocrine IGF-II /IRA Signal In Malignant Mesothelioma' sought to elucidate the mechanism of EphB4 stabilization via IGF-II. The authors performed a very clear study and designed the experimental procedure in a very appropriate way. In the end, they identified a new DTX3 E3 ligase isoform (DTX3c). This study should be included in the literature. I have very small suggestions.

-In Figure 1C, the colors are very similar to each other, which makes it difficult to distinguish the data. It would be better to write the percentages to the identifications to make them clearer.

-Can the authors provide any enzymatic assay data to confirm inhibition of the proteasome by MG132 and of cdc48/p97 by ML240?

Author Response

The manuscript titled 'Novel Isoform DTX3c Associates With UBE2N-UBA1 And Cdc48/p97 As Part Of The EphB4 Degradation Complex Regulated By The Autocrine IGF-II /IRA Signal In Malignant Mesothelioma' sought to elucidate the mechanism of EphB4 stabilization via IGF-II. The authors performed a very clear study and designed the experimental procedure in a very appropriate way. In the end, they identified a new DTX3 E3 ligase isoform (DTX3c). This study should be included in the literature. I have very small suggestions.

- In Figure 1C, the colors are very similar to each other, which makes it difficult to distinguish the data.

  It would be better to write the percentages to the identifications to make them clearer.

Authors reply: The size of picture 1C was increased in order to better identify colors and percentages

- Can the authors provide any enzymatic assay data to confirm inhibition of the proteasome by MG132   

   and of cdc48/p97 by ML240? Authors reply: Rational for the use of inhibitors at the concentration shown in the figure has been further substantiated both in the figure legend and under Methods.

Reviewer 2 Report

I suggest to improve the presentation and quality of the images. For example in Figure 2, a and b panels are moved. Also in figure 3, panels in a and b should be formated in a similar way.

In Figure 4, please define statistical significance of * and **.

Author Response

I suggest to improve the presentation and quality of the images. For example in Figure 2, a and b panels are moved. Authors reply: Figures 2a and b have been reformatted to show consistency in size and fonts. Also in figure 3, panels in a and b should be formatted in a similar way.

Authors Reply: Done

In Figure 4, please define statistical significance of * (p<0.05) and ** (p<0.01). Authors Reply: The significance of the individual asterisks was specified in the figure.

Reviewer 3 Report

In the study ''Novel Isoform DTX3c Associates With UBE2N-UBA1 And 2 Cdc48/p97 As Part of The EphB4 Degradation Complex Regulated By The Autocrine IGF-II/IRA Signal In Malignant Mesothelioma'' the authors use a combination of targeted proteomics and an in house developed biotin captured assay to identify components of the ubiquitin proteasome system (UPS) that interact with the  EphB4 kinase under conditions of IGF2 dependent autocrine stimulation. The studies on EphB4 are warranted since EphB4 is reported in mesothelioma. Overall this is  a solid study that utilizes multiple techniques to identify interacting partners. In addition the authors identify a previously undescribed isoform of the DTX3 E3 ubiquitin ligase (DTX3c) that has altered structural properties and may have distinct functions in mesothelioma.  While the study is of interest I think that the lack of additional mesotheliama cell lines is a weakness. Perhaps the authors should refer to why the MSTO211H is the chosen cell line and if they expect that their findings will be applicable in other mesothelioma cell lines. In addition the blots in fig 2 are difficult to interpret (presumably due to the difficulties in isolating the EphB4 degradation complex). 

Author Response

In the study ''Novel Isoform DTX3c Associates With UBE2N-UBA1 And 2 Cdc48/p97 As Part of The EphB4 Degradation Complex Regulated By The Autocrine IGF-II/IRA Signal In Malignant Mesothelioma'' the authors use a combination of targeted proteomics and an in house developed biotin captured assay to identify components of the ubiquitin proteasome system (UPS) that interact with the  EphB4 kinase under conditions of IGF2 dependent autocrine stimulation. The studies on EphB4 are warranted since EphB4 is reported in mesothelioma. Overall this is  a solid study that utilizes multiple techniques to identify interacting partners. In addition the authors identify a previously undescribed isoform of the DTX3 E3 ubiquitin ligase (DTX3c) that has altered structural properties and may have distinct functions in mesothelioma.  While the study is of interest I think that the lack of additional mesothelioma cell lines is a weakness. Perhaps the authors should refer to why the MSTO211H is the chosen cell line and if they expect that their findings will be applicable in other mesothelioma cell lines.

Authors reply: “The reasoning for referring to “Mesothelioma” in the text rather than to th eindividual cell line relates to the previously published findings in mesothelioma cell lines recapitulating the three known histological cancer types conveyed in Scalia et al Oncogene 2019 which clearly show the presence of the autocrine IGF-II/IR-A->EphB4 regulatory axis in the studied mesothelioma cell lines which histotypes span from the epithelial to the anaplastic phenotype. We showed that the response to the IGF-II block is higher in the most aggressive cell histotypes (MSTO211H and 2025H and lesser in H2028). MSTO211H (high malignant grade) in the above study, was the most responsive mesothelioma cell line to the EphB4 protein degradation rescue control by the IGF-II loop, which guided our choice and research design conveyed in the present study in order to pinpoint the underlying IGF-II-regulated degradation machinery. This concept has been reinforced in the text under the “Abstract”, “Introduction” and Results sections.

In addition the blots in fig 2 are difficult to interpret (presumably due to the difficulties in isolating the EphB4 degradation complex).

Authors reply: The interactions displayed in Figure 2 relate to rare protein events occurring on a limited protein pool within the studied cell model which motivated us to improve the traditional co-immune precipitation approach (not shown) by including PTM enhancement factors in the displayed pull-down approach by using an equal and consistent amount of EphB4 phosphodegron synthetic peptide bait between treated and control conditions. Ultimately is was possible to unmask the effect of the IGF-II signal deprivation on the recruitment of the individual degradation components confirming their specific involvement in the previously identified biological process.